# The Effect of Environmental Factors on Immunological Pathways of Asthma in Children of the Polish Mother and Child Cohort Study

**DOI:** 10.3390/ijerph20064774

**Published:** 2023-03-08

**Authors:** Żywiołowska-Smuga Sara, Jerzyńska Joanna, Podlecka Daniela, Polańska Kinga, Brzozowska Agnieszka

**Affiliations:** 1Department of Pediatrics and Allergy, Copernicus Memorial Hospital, Medical University of Lodz, Pabianicka Street 62, 93-513 Lodz, Poland; 2Department of Environmental and Occupational Health Hazards, Nofer Institute of Occupational Medicine, 91-348 Lodz, Poland

**Keywords:** asthma, children, regulatory T cells, FOXP3, environmental factors

## Abstract

The FOXP3 transcription factor is a marker of regulatory T cells (Tregs), and is essential in the process of their activation and proper expression by promoting immune homeostasis. To assess the influence of the environment on the development of asthma, we hypothesized that in our cohort, exposure to environmental factors is associated with asthma risk in children, and that FOXP3 levels vary with their incidence and are negatively correlated with developing asthma. This prospective study conducted in Poland uses a cohort of 85 children (42 with and 43 without asthma diagnosis) aged 9 to 12 years recruited for the Polish Mother and Child Cohort Study. We collected questionnaires and organized visits to assess patients’ clinical condition (skin prick tests, lung function assessments). Blood samples were taken to determine immune parameters. Breastfed children had lower risk of asthma. Asthma risk was higher in children who live in the city, with antibiotic course before the age of 2 and antibiotic therapy more than twice a year. Environmental factors were associated with childhood asthma. Breastfeeding, the coexistence of other allergic diseases, and the frequency of housekeeping affect FOXP3 levels, which are negatively correlated with the risk of asthma.

## 1. Introduction

Recent decades have brought a marked increase in the incidence of asthma, although the knowledge about its proper diagnosis and treatment has also increased. The etiology of asthma is complex. Factors that may increase the risk of developing childhood asthma have been identified, such as exposure to allergens, respiratory tract infections, frequent use of antibiotics, living conditions, and exposure to tobacco smoke [1]. The development of urbanization and its consequences, such as the loss of biodiversity in general, critically affects human immune homeostasis [2,3].

We know little about the immunological mechanisms by which asthma develops into a persistent disease, or by which symptoms regress. For example, the impact of environmental factors on immunoregulation processes in childhood asthma is remains to be understood. The induction and maintenance of tolerance is broadly influenced by a wide range of cell types and suppressive molecules. Immune responses are controlled by a heterogeneous subpopulation of lymphocytes—T regulatory cells (Tregs) [4]. By acting on the effector cells of the inflammatory reaction, Tregs suppress the excessively high or autoreactive inflammatory response. Their action is crucial to maintain the immune balance and tolerance to own antigens [5]. The transcription factor FOXP3 is a characteristic marker of Treg cells and is essential for their activation, proper development, and functioning [6]. Appropriate expression of FOXP3 modifies the allergen-specific Th2 lymphocyte response in favor of Th1 by various mechanisms and supports the immune tolerance [7]. The maintenance of the stability of Tregs is subject to epigenetic regulation, which consists of potentially inherited changes in gene expression that are influenced by the environment. The consequences of environmental exposure, such as DNA hypermethylation, histone modifications, and post-transcriptional modifications, are regulation of FOXP3 expression, which influences the distribution of Tregs CD4+ CD25+ FOXP3+, and certain epigenetic changes are significantly associated with asthma [8,9]. The increased expression of IL-2 receptor (CD25) may reflect an increased activation state [10,11,12]. Furthermore, the activation of transcription factors, such as the suppressor of cytokine signaling (SOCS) and peroxisome proliferator-activated receptor gamma (PPARγ), may also influence the course of inflammation (PPARγ brings an anti-inflammatory effect) [11,13]. Various cytokines, including IL-6 and IL-10, induce the expression of cytokine signaling 3 (SOCS3) [14,15]. Dominant repeats of glycoprotein A (GARP) are one of the elements building peripheral tolerance (activation of TGF-β1) [16,17].

This study was designed to investigate the clinical and immunological parameters in children aged 9–12 years old from the previously formed Polish Mother and Child Cohort (REPRO_PL), comparing patients with and without asthma, taking into account the exposure of the examined groups to various environmental factors. We expected that the expression of FOXP3 would be lower in asthma patients and that the data collected on exposure to environmental factors in these patients will also be significantly different in the mentioned groups.

## 2. Materials and Methods

### 2.1. Participants

This is a prospective study based on data of children from the Polish Mother and Child Cohort (REPRO_PL). REPRO_PL was established in 2007 in Poland [18,19,20]. The recruitment process and follow-up procedures, together with a complete description of the methodological assumptions, have been published elsewhere [21,22]. Briefly, in 2007–2011, 1300 mother–child pairs were recruited during the first trimester of pregnancy at maternity units or clinics in two regions of Poland (Lodz and Legnica) if they fulfilled the following inclusion criteria: single pregnancy up to 12 weeks of gestation, no assisted conception, no pregnancy complications, and no chronic diseases, as specified in the study protocol [21]. When children of mothers participating in the REPRO_PL Cohort were 9–12 years old, information was sent to parents inviting them to participate in the continuation of the research. At this stage, 250 mother–child pairs agreed to continue to participate in the study. The current analysis counts 85 patients at the age 9–12 years who have completed three scheduled visits of the study, as described below. A study group included 42 patients diagnosed with asthma (all cases of diagnosed asthma in the whole cohort), and a control group included 43 patients without asthma diagnosis for comparison. All mothers gave their written consent to take part before the follow-up study. Approval was given by the Ethical Committee of the Nofer Institute of Occupational Medicine, Lodz, Poland (Decision No. 7/2007, 3/2008, and 22/2014), for the primary study and by the Medical Ethics Committee of the Medical University of Lodz (Decision No. RNN/388/17/KE) for the follow–up study.

### 2.2. Study Design

Between nine and twelve years after the birth, an invitation letter was sent to mothers with an invitation to participate in follow-up examinations covering exposure and an assessment of the child’s health status. Parents were interviewed to collect demographic and socioeconomic data, and a medical and reproductive history. The data was recorded between January 2019 and November 2020.

In order to collect questionnaire data and to reliably assess the patient’s clinical condition and immune profile, individual visits to the ward were arranged. At the first visit, the questionnaire data was collected and skin prick tests (SPT), or specific IgE (where it was not possible to perform the SPT) to assess the allergy status, were performed. On scheduled visits, the patients appeared in general health, with no signs of infection. Another visit was arranged 6 months later, and a final third visit another 6 months later. The second visit focused on pulmonary function assessment by spirometry and exhaled NO testing. The third visit was to control asthma using the previously used tools and, additionally, to collect blood to determine the immune parameters. Group membership was verified after each visit. Detailed information on the methods employed is given elsewhere [23].

### 2.3. Child Sociodemographic, Exposure, and Health Assessment

The questionnaire provided knowledge on sociodemographic data, mode of delivery, birth weight, breastfeeding, dampness, carpets surface, frequency of cleaning, having pets at home, and exposure to nicotine smoke. Health status was assessed in children between 9 and 12 years of age. Briefly, a questionnaire was administered to the mothers, and this was supplemented with information from the medical chart of each child. This part of the questionnaire has been developed by an allergist, based on recommendations from the International Study of Asthma and Allergies in Childhood (ISAAC), and has been applied previously [22,24]. In addition, the occurrence of allergies among family members was noted. A clinical examination was performed by a pediatrician/allergist in the presence of the mother or a relative. Patients of the study group were defined as having asthma, allergic rhinitis, or atopic dermatitis if they had ever been diagnosed by a physician. Pediatric asthma was diagnosed by a specialist in accordance with the recommendations of GINA (Global Initiative for Asthma) [25], which defines asthma as a heterogeneous disease characterized by chronic inflammation of the airways, the symptoms of which include wheezing, coughing, shortness of breath, and chest tightness. It has been taken into account that, as reported by GINA, symptoms vary over time and have varying degrees of severity, and that variable expiratory airflow limitations may coexist, which may become permanent over time. The diagnosis was additionally supported by pulmonary function tests, an improvement after dedicated asthma treatment, or symptoms indicative of bronchial hyperresponsiveness, such as a tendency for excessive contraction in response to stimuli such as physical or chemical factors that do not cause such an effect in a healthy person. Patients from the control group, which included children who never presented with asthma symptoms, confirmed group membership in questionnaire data and clinical tests.

### 2.4. Allergen Sensitization

Skin prick testing was performed using standard allergen extracts from Allergopharma (Reinbek, Germany). A reaction >3 mm in diameter above the negative control recorded at 15 min were considered positive. Allergen sensitization was defined as a specific IgE of ≥0.35 KU/L for at least one of tested allergens (chemiluminescence method (CLIA), Immulite 2000, XPI, Siemens, Munich, Germany).

### 2.5. Lung Function Assessment

All pulmonary function tests were performed with a Master Screen unit (Erich Jaeger Gmbh-Hochberg, Friedberg, Germany), as described elsewhere, in accordance with the ATS/ERS guidelines [26]. After three measurements, the highest FEV1 value was taken from the three performed trials. Parameters, such as FEV1, air volume during the first second of forced exhalation (FEC1%)/FVC (Tiffneau index) was assessed. The FENO test was performed using the chemiluminescence method using an analyzer from Sievers Instruments Inc, Boulder, CO, USA, model 280i. As recommended by ATS, the test was performed prior to spirometric maneuvers to avoid false-low results. The expiratory flow rate was kept constant at 50 mL/s for at least 3 s throughout the study. The result was given in ppb units as an averaged value from 3 correctly performed measurements.

### 2.6. Immunological Assessment

To evaluate the immunological parameters, 5 mL of peripheral blood was collected from each patient. The assessment of immunological parameters was performed using the multicolor flow cytometry technique. A panel of antibodies conjugated to isothiocyanate fluorescein (FITC), phycoerythrin (PE), Peridinin-chlorophyll-protein (Per-CP), or allophytocyanin (APC) was used for the determinations. All procedures were performed according to the manufacturer’s instructions. The following were determined by flow cytometry: CD4+/CD25+, CD25+/CD71+, FOXP3+, GARP+, PPAR+/11c+, and SOCS3.

### 2.7. Statistical Analysis

Statistical calculations were computed with R software (R-4.1.2.). The description of categorical traits consisted of the number of observations with their percentage share in each respective group. The description of numerical variables was based on adequate statistics, depending on the normality of distribution. Comparison of asthma vs. non-asthma groups was run with an independent Student’s *t*-test, Mann–Whitney U test, Chi-squared test, or Fisher’s exact test, as appropriate. Normality was checked with the Shapiro–Wilk test, accompanied with skewness and kurtosis parameters. Variance homogeneity was verified with the Levene test. Logistic regression analysis consisted of univariate logistic regression models for each available variable. All tests assumed alpha = 0.05 significance.

## 3. Results

Eighty-five patients were included into the analysis: 42 with asthma (the study group) and 43 without asthma (the control group). Clinical characteristics of study groups are presented in Table 1.

### 3.1. Univariate Logistic Regression

Logistic regression analysis was conducted to understand which factors influenced the risk of asthma in statistically significant way. Due to the number of observations, univariate models were run for each available variable. Results are presented in Table 2.

#### 3.1.1. Environmental Factors and Asthma Risk

In case of using forceps or vacuum during birth, the odds of asthma were nearly nine-fold higher, compared to natural spontaneous delivery, OR = 8.61 CI_95_ [2.09;58.79], *p* = 0.008. For patients who were breastfed, the asthma risk turned out to be 96% lower than for the others, OR = 0.04 CI_95_ [0.00;0.21], *p* = 0.002. City residence was associated with seven times higher asthma risk, OR = 7.22 CI_95_ [2.13;33.40], *p* = 0.004, while with village residence location, the odds were lower by 87%, OR = 0.13 CI_95_ [0.03;0.44], *p* = 0.003. For patients having carpets at home, the risk of asthma was six-fold higher than for those without carpets, OR = 6.21 CI_95_ [2.03;23.52], *p* = 0.003. Antibiotic variables turned out to be significant for the level of asthma risk. Taking antibiotics up to the age of 2 years was associated with a seven-fold higher asthma risk, OR = 7.33 CI_95_ [2.85;20.40], *p* < 0.001, while with antibiotic therapy used more than twice annually, the asthma risk was six times higher, OR = 6.40 CI_95_ [2.53;17.35], *p* < 0.001.

#### 3.1.2. Family History and Asthma Risk

An almost sixteen times higher risk of asthma was observed for children with already diagnosed allergies in families, OR = 15.58 CI_95_ [4.71;71.65], *p* < 0.001. Allergic rhinitis (AR)/conjunctivitis, atopy, HDM allergy, allergy to molds, and allergy to grass/trees pollen were associated with higher odds of asthma in children (OR = 38.06 CI_95_ [12.02;146.10], *p* < 0.001, OR = 102.86 CI_95_ [24.43;733.27], *p* < 0.001, OR = 14.67 CI_95_ [4.43;67.36], *p* < 0.001, OR = 5.59 CI_95_ [1.33;38.36], *p* = 0.035, OR = 24.32 CI_95_ [8.11;87.00], *p* < 0.001, respectively).

#### 3.1.3. Immunological Parameters and Asthma Risk

FOXP3 levels were found in the blood of patients in both groups. The level of FOXP3 was negatively related to the risk of asthma; its level higher by one unit was connected with the risk lowered by 5%, OR = 0.95 CI95 [0.93;0.97], *p* < 0.001. No association was found between asthma and other immunological parameters we considered in our study.

#### 3.1.4. Lung Function Test Results and Asthma Risk

An FeNO level higher by one ppb corresponded to an increased risk of asthma by 12%, OR = 1.12 CI_95_ [1.04;1.23], *p* = 0.010. Logistic regression results showed that all FEV-related parameters had an impact on asthma risk; however, despite statistical significance, the FEV conclusions were not consistent.

### 3.2. Multivariate Logistic Regression

Further, multivariate logistic regression analysis was conducted. The first step Directed Acyclic Graph (DAG) was used to model a priori causal assumptions and inform variable selection strategies for causal questions (Figure 1). Only these factors that were directly related to the outcome and these for which the sufficient number of participants in the study groups were considered. Due to the total number of observations, the multivariate model had no more than five explanatory variables. Common factors suspected to have a medical importance, whose asthma risk assessment would be valuable to many were selected, included breastfeeding, city residence, coexistence of other allergic diseases, and use of antibiotics in early childhood, which were relatively known potential factors playing a role when asthma is considered. However, we also wanted to investigate FOX3. The results of our research showed that environmental factors can influence the level of FOXP3, which in turn leads to an increased risk of asthma. We found this hypothesis interesting and wanted to see if the multivariate model would confirm these assumptions. In addition, the choice was supported by reports from previous studies by other authors and the results of the significance *p* of the univariate model of our study. Final components of the multivariate model were the result of AIC (Akaike Information Criterion) verification and VIF (Variance Inflation Factor) indicators assessment. The model was run for FOXP3, breastfeeding, use of antibiotic therapy up to the age of 2 years, other allergic diseases, and patients’ residence (city/village). The results of the analysis are presented in Table 3.

Significant importance for asthma risk was identified for the prevalence of other allergic diseases, exposition on antibiotics up to 2 years, and FOXP3. For breastfeeding and city residence, a significant relationship with asthma risk was not confirmed (*p* = 0.204 and *p* = 0.274, respectively). Other allergic diseases would relate to and increased asthma risk by 41 times (*p* = 0.002). Exposition to antibiotics up to the age of 2 years was associated with risk level seven times higher (*p* = 0.019), and with FOXP3 higher by one, a 6% lower risk of asthma was observed (*p* = 0.001). Model verification was based on Nagelkerke R2, which was equal to 76.3% and Hosmer and Lemeshow GOF analysis (*p* = 0.988). Both indicators confirmed robustness of the model.

### 3.3. FOXP3 Level Depending on Selected Environmental Factors

We hypothesized whether environmental factors can affect FOXP3 levels and thus have an impact on the risk of developing asthma. To develop this hypothesis, the level of FOXP3 was analyzed in terms of selected environmental factors, such as breastfeeding, place of residence (city/village), exposure to antibiotics before the age of 2 years, presence of antibiotics at least twice a year, coexistence of other allergic diseases, and frequency of cleaning at home. As the FOXP3 distribution was found not to be consistent with the normal distribution, the verification of the difference in the levels of this variable depending on the examined factors was performed using non-parametric statistical tests. The results of this analysis are presented in Table 4.

The results showed that the level of FOXP3 was significantly dependent on breastfeeding, coexistence of allergic diseases, and the frequency of cleaning at home. Breastfeeding was associated with a higher level of FOXP3 (median = 49.60 vs. median = 21.50 in non-breastfed patients), *p* = 0.001. The coexistence of allergic disease was associated with a lower level of FOXP3 (median = 32.70 vs. median = 58.60 in patients without other allergic diseases), *p* = 0.001. However, in the case of cleaning, the highest level of the FOXP3 index was found in the group where cleaning was performed most often (more than twice a week), median = 57.70, and the lowest FOXP3 level was found in people whose house was cleaned twice a week, median = 34.90 (*p* = 0.043). The post-hoc test showed that the pair responsible for the significance of the relationship between the frequency of cleaning and the FOXP3 level was the group with cleaning more than twice a week and the group with cleaning twice a week (*p* = 0.014). In the case of the other examined factors, the relationship between the level of FOXP3 and these factors was not statistically confirmed (*p* > 0.05).

### 3.4. ROC Curve

Diagnosis of asthma must be based on standard definitions; however, certain tools can help predict the risk of developing asthma. We assessed the predictive value of FOXP3 in detecting asthma by analyzing the ROC curve and its accompanying indices. Figure 2 presents the AUC (area under the ROC curve), which was calculated and used to assess the predictive value of the test (maximum AUC is 1). In the case of the test, AUC equal to 0.821 indicated a high predictive value of the analyzed test.

The ROC analysis results are presented in Table 5. The sensitivity of the asthma predictive test constructed on the basis of FOXP3 was 67%, specificity was 86%, and the accuracy was 76%.

## 4. Discussion

The mechanisms leading to the development of asthma in children are still under investigation. The respiratory tract is the habitat of many bacteria and fungi forming a complex network, which is beneficial for the human organism as long as there is a balance between the elements of the respiratory microbiome. This local beneficial flora of the respiratory system enhances complex immune mechanisms; however, factors such as pathogens, antibiotic therapy, and medical interventions, such as cesarean section, disturb the microbiome homeostasis. The modifications of the microflora include the multiplication of pathogenic bacteria or their movement to other locations of the system, which promote the development of chronic diseases such as asthma [27]. The role of the environment in the development of the immune system is undeniable, and the exposure to certain factors, especially in the early stages of a child’s life, can have both protective and negative effects.

The diagnosis of the asthma epidemic in recent years has led to the creation of series of cohorts, which also confirmed that the environmental factors discussed in this study contribute to the development of asthma. So far, however, little is known, and it is difficult to accurately assess the individual risk factors and their mechanisms.

Our study analyzed various environmental factors in relation to the risk of developing asthma as well as the level of FOXP3 in relation to selected factors and examined how it changes depending on their occurrence. We proved that the level of FOXP3 was significantly dependent on breastfeeding. The benefits of natural infant feeding, such as the immunostimulatory effects of IgA, polysaccharides, cytokines, and proteins in breast milk [28], are well known and it has also been proven that extending breastfeeding by one month can reduce the risk of asthma by 23% [29]. Our study group hypothesized that one of the mechanisms of this antiallergic defense and the maintenance of immune homeostasis is the action of FOXP3, as the breastfed infants had higher levels of this T-cell transcription factor.

Various studies have looked at altering FOXP3 levels in asthma. For example, a Chinese study conducted on a small group of asthmatic patients with and without asthma showed decreased CD4+ CD25+ FOXP3+ Treg levels and lower FOXP3 mRNA expression in asthmatic patients compared to healthy children. In the same study, this relationship was also evident in the positive correlation between the level of FOXP3 mRNA expression and the results of lung function tests, such as FEV1 [30]. Our observations showed, however, that the level of FOXP3 was also influenced by the coexistence of other allergic diseases (at least one of the following: atopic dermatitis, food allergy, allergic rhinitis, or atopy), and the difference was statistically significant. This confirmed the importance of the burden of other allergic diseases, as the multifactorial logistic regression model also unequivocally showed that the coexistence of other allergic diseases may accompany the development of asthma.

Keeping cleanliness in a home environment where the youngest spend a lot of time is important in the aspect of allergies. Removing dirt or dust plays a significant role in the development of asthma. A group of Polish researchers, on the basis of house dust samples, proved the impact of the density and species composition of house mites in apartments on the severity of bronchial asthma in the studied children [31]. They also proved that the presence of domestic animals has a positive effect on the density of domestic mites, which we were not able to confirm in our group in a logistic regression analysis. We focused on a frequency of cleaning and the results clearly showed that the children who live in homes cleaned more than twice a week have higher FOXP3 levels than other children.

Many studies have emphasized that correct FOXP3 expression plays a key role in maintaining tolerance to autoantigens and induced Treg lymphocytes, which is in line with our assessment. Chronic allergen-specific immunotherapy in murine models highlights the modulating role of this T-cell population. Böhm et al. showed that such therapy, by inducing Treg CD4 + FOXP3+ lymphocytes, normalizes the balance between Tregs and other T cell populations and increases the synthesis of anti-inflammatory IL-10 in the lungs [32]. Moreover, it is known that subcutaneous injection of an allergen extract or administration of the preparation by sublingual or oral route modifies the allergen-specific Th2 lymphocyte response in favor of Th1, which generally protects against allergies.

Adverse environmental exposure can significantly affect the Treg cell population and contribute to FOXP3 hypermethylation. It is commonly believed that the air in the cities is of lower quality than in villages. Although, in our study, we did not find statistically significant differences between the levels of FOXP3 in children from rural and urban areas, this result could be due to the imperfect differentiation of patients in the study and control groups, because the proportions of children living in the countryside to those living in the city were not equal. However, one study has shown that environmental-induced epigenetic changes are significantly associated with asthma. Prunicki et al. proved a positive relationship between the level of methylation of the FOXP3 promoter and exposure to air pollution and also linked asthma with regional differences in the methylation of the FOXP3 promoter region [33]. They also proved that these changes persisted over time. Similarly, it was confirmed in another study that processes, such as the mentioned DNA methylation, histone modifications, and post-transcriptional modifications, regulate the expression of FOXP3 and influence the distribution of Treg CD4+ CD25+ FOXP3+ [34].

It has been proved that both prenatal and postnatal antibiotic exposure have been associated with an increased risk of asthma [35,36]. In a cohort study of over 80 thousand mothers and their children, it was found that the use of broad-spectrum antibiotics was associated with an increased likelihood of developing asthma in the child, and that the increased cumulative dose increased the risk of asthma [37]. Another study showed that the risk of asthma increases with each subsequent course of maternal antibiotic therapy during pregnancy, and commonly used antibiotics, such as β-lactam penicillin, macrolides, and lincosamides, were indicated among the antibiotics associated with childhood asthma [38]. In our study, we found that the use of antibiotics before the age of 2 years had a significant impact on the development of asthma, and the use of more than two antibiotics per year in the studied population was associated with a higher incidence of asthma. When considering the effect of antibiotic therapy before the age of 2 years on FOXP3 levels, we found no statistically significant correlation.

This study has some limitations. We gathered a group of children of different ages, and therefore, with different maturity of the immune system. It is known that the levels of immune markers change with the age of the patient, and even healthy subjects can be slightly abnormal. We are aware that asthma is a heterogeneous disease with different endotypes and phenotypes, and due to diagnostic difficulties in children, some of our patients may not have been properly diagnosed or have not developed symptoms when entering the study. However, our goal was not to differentiate endo- and phenotypes of asthma, but to raise awareness of the importance of the environmental factors influencing the development of the disease. Further investigation in a much larger group of children is needed to show the correlation of the factors discussed in our study with immunity parameters more precisely.

This study is the first to so clearly show that FOXP3 levels fluctuate significantly depending on a variety of environmental factors associated with asthma. A detailed assessment of FOXP3 levels was neither the main nor the most important original objective of the study. However, we hypothesized whether environmental factors affecting FOXP3 levels may indirectly affect asthma risk. Taking into account the obtained results of the analysis shown in the paper (Table 4; Level of FOXP3+ depending on selected environmental factors), we concluded that it can be suspected that environmental factors, both protective ones that increase the level of FOXP3, as in the case of breastfeeding or more frequent cleaning at home, as well as the negative ones that lower the level of FOXP3, as in the presence of other allergic diseases, may affect the risk of asthma, respectively, protectively or by causing the development of the disease. This hypothesis requires further research. Observing how environmental factors change selected immunological parameters may be a valuable step towards new diagnostic and therapeutic tools in asthma. We also emphasized that FOXP3 is a parameter of high predictive value with a specificity of 86% and a sensitivity of 76%, and these results we found to be highly satisfactory. 

Due to the limited number of participants in the study and multiple analyses performed based on the group, there are odds of finding false positive effects. Therefore, the analysis is suggested to be treated in an explanatory manner. The outcomes play the role of hypotheses at this stage, and we suggest further exploration within an extended sample of participants.

Taking into account the young age of our patients, our results suggest that the development of asthma is gradual from early childhood and is closely related to immunoregulation. Future research should bring together groups of pediatric and adult patients, also taking into account the difficult group of young children, where diagnosis of asthma is a clinical challenge. The study of the phenotypes and endotypes of allergic diseases enables the effective identification of the pathomechanism of the disease and is extremely important in the work on targeted biological therapies. It is possible that the deeper knowledge of immune biomarkers, such as FOXP3, its changes in expressions, and modifications, as well as the changes of the expression of other parameters related to asthma, will enable new targeted therapies in the future and provide, in addition to anti-inflammatory drugs, new pillars of asthma therapy.

## 5. Conclusions

In conclusion, our study found that the environmental factors are associated with childhood asthma. Protective factors in childhood asthma are breastfeeding and living in the countryside. Factors that may be associated with a higher risk of developing asthma in children include family allergies, co-occurrence of another allergic disease, use of antibiotics up to the age of 2 years or more often than twice a year, and living in a city. Breastfeeding, the coexistence of other allergic diseases, and the frequency of housekeeping all affect FOXP3 levels, which are negatively correlated with the risk of asthma. Further study is needed to gather information on this interaction between environmental factors and the immune system in prevention and treatment of asthma.

## Figures and Tables

**Figure 1 ijerph-20-04774-f001:**
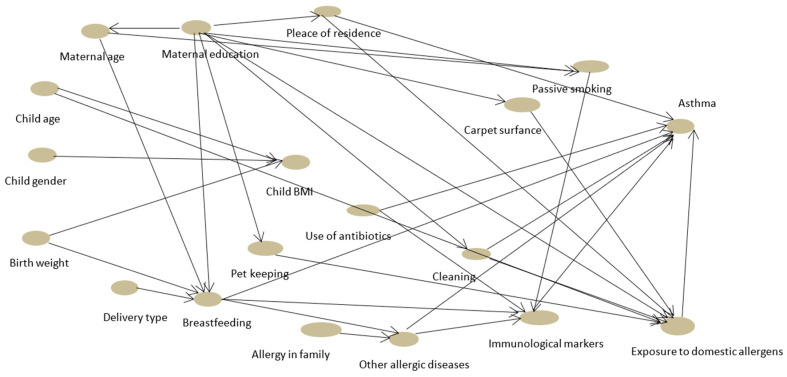
The covariates for multivariate logistic regression.

**Figure 2 ijerph-20-04774-f002:**
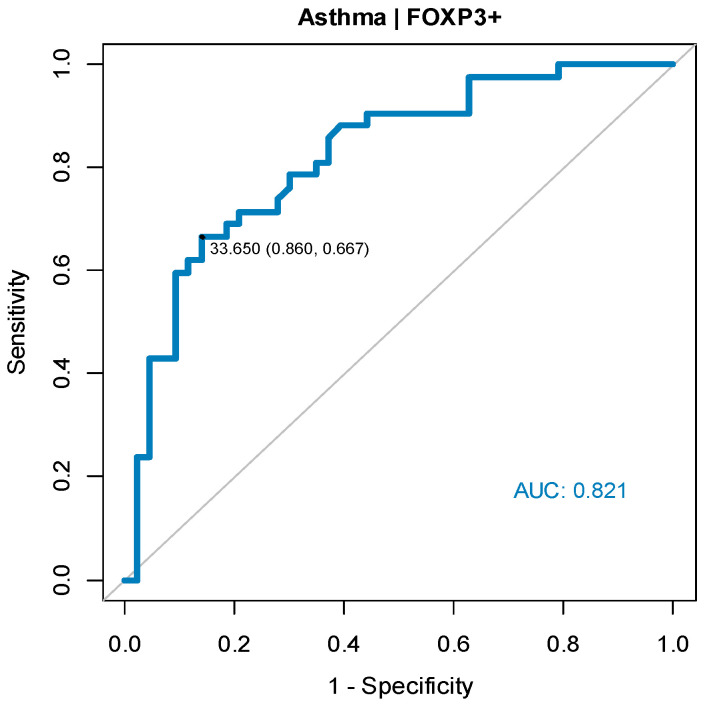
ROC curve predicting asthma risk based on FOXP3 level.

**Table 1 ijerph-20-04774-t001:** Characteristics of the study group and control group.

Variable	Total (N = 85)	Asthma (N = 42)	No Asthma (N = 43)
Gender, n (%)			
Female	42 (49.4)	20 (47.6)	22 (51.2)
Male	43 (50.6)	22 (52.4)	21 (48.8)
Age, n (%)			
9 years	21 (24.7)	5 (11.9)	16 (37.2)
10 years	20 (23.5)	6 (14.3)	14 (32.6)
11 years	22 (25.9)	14 (33.3)	8 (18.6)
12 years	22 (25.9)	17 (40.5)	5 (11.6)
BMI, kg/m^2^, median (Q1; Q3)	19.00 (17.00; 20.00)	19.00 (17.00; 21.00)	19.00 (18.00; 19.00)
Birth weight, g, mean ± SD	3324.88 ± 332.05	3357.50 ± 316.37	3293.02 ± 347.43
Mother’s age, years, mean ± SD	28.37 ± 3.86	29.20 ± 3.73	27.58 ± 3.86
Delivery type, n (%)			
Natural spontaneous	56 (66.7)	23 (54.8)	33 (78.6)
Natural induced	9 (10.7)	5 (11.9)	4 (9.5)
Forceps or vacuum	14 (16.7)	12 (28.6)	2 (4.8)
Cesarean section	5 (6.0)	2 (4.8)	3 (7.1)
Breastfeeding *, n (%)	68 (80.0)	26 (61.9)	42 (97.7)
Mother’s education, n (%)			
Secondary	16 (18.8)	6 (14.3)	10 (23.3)
Higher	69 (81.2)	36 (85.7)	33 (76.7)
Place of residence			
City, n (%)	66 (78.6)	39 (92.9)	27 (64.3)
Village, n (%)	19 (22.4)	3 (7.1)	16 (37.2)
Allergy in family, n (%)	56 (67.5)	37 (92.5)	19 (44.2)
Atopic dermatitis, n (%)	16 (18.8)	11 (26.2)	5 (11.6)
Food allergy, n (%)	15 (17.6)	10 (23.8)	5 (11.6)
AR/conjunctivitis, n (%)	44 (51.8)	37 (88.1)	7 (16.3)
Atopy, n (%)	47 (55.3)	40 (95.2)	7 (16.3)
Type of allergy			
HDM, n (%)	25 (29.4)	22 (52.4)	3 (7.0)
molds, n (%)	11 (12.9)	9 (21.4)	2 (4.7)
grass/trees pollen, n (%)	37 (43.5)	32 (76.2)	5 (11.6)
dog/cat, n (%)	10 (11.9)	6 (14.3)	4 (9.5)
Household exposure			
Nicotine exposure, n (%)	14 (16.5)	6 (14.3)	8 (18.6)
Domestic animals, n (%)	38 (44.7)	20 (47.6)	18 (41.9)
Dampness, n (%)	26 (30.6)	15 (35.7)	11 (25.6)
Carpet surface, n (%)	64 (75.3)	38 (90.5)	26 (60.5)
Cleaning, n (%)			
Less than once a week	1 (1.2)	0 (0.0)	1 (2.3)
Once a week	20 (23.5)	11 (26.2)	9 (20.9)
Twice a week	42 (49.4)	25 (59.5)	17 (39.5)
More than twice a week	22 (25.9)	6 (14.3)	16 (37.2)
Use of antibiotics up to the age of 2, n (%)	47 (56.0)	33 (78.6)	14 (33.3)
Use of antibiotics more than 2 per year in childhood, n (%)	46 (54.8)	32 (76.2)	14 (33.3)
Immunological markers			
PPARG+/11c+, median (Q1; Q3)	12.50 (5.80; 21.80)	14.55 (8.20; 21.87)	10.10 (4.25; 19.35)
25+/4+, median (Q1; Q3)	4.50 (2.10; 10.40)	7.50 (2.97; 12.45)	3.50 (1.80; 6.40)
FOXP3+, median (Q1; Q3)	43.40 (22.40; 68.40)	24.75 (14.95; 43.93)	60.80 (41.20; 90.55)
GARP+ (I), median (Q1; Q3)	45.20 (22.20; 87.40)	43.85 (16.90; 91.10)	52.40 (34.50; 81.60)
25+/71+ (I), median (Q1; Q3)	4.10 (2.60; 6.70)	3.95 (2.80; 6.20)	4.20 (2.25; 7.25)
SOCS+, median (Q1; Q3)	18.10 (7.30; 36.10)	26.60 (14.75; 42.55)	11.30 (5.80; 28.10)
25+/71+ (II), median (Q1; Q3)	4.30 (2.30; 7.40)	4.35 (3.20; 6.57)	3.70 (2.00; 10.10)
GARP+ (II), median (Q1; Q3)	15.20 (8.00; 28.50)	14.10 (8.83; 24.82)	16.00 (7.55; 36.20)
FEV1, L, median (Q1; Q3)	1.90 (1.60; 2.54)	2.29 (1.68; 3.00)	1.76 (1.58; 2.02)
FEV1, %, median (Q1; Q3)	89.00 (84.00; 96.00)	87.00 (81.50; 90.00)	92.00 (88.50; 98.00)
FEV1/FVC, %, mean ± SD	0.91 ± 0.11	0.88 ± 0.10	0.95 ± 0.10
FeNO, ppb, median (Q1; Q3)	28.00 (17.40; 35.40)	32.40 (22.35; 43.65)	16.25 (13.83; 24.68)

BMI—body mass index, AR—allergic rhinitis, HDM—house dust mites, PPARG+—Peroxisome proliferator-activated receptor gamma, FOXP3+—Forkhead box P3, GARP+—Glycoprotein A repetitions predominant, SOCS+—Suppressors of cytokine signaling, FEV1—forced expiratory volume in first second, FVC—forced vital capacity, FEV1/FVC ratio—also called Tiffeneau–Pinelli index, the ratio of the forced expiratory volume in one second to the full forced vital capacity, FeNO—fraction of exhaled nitric oxide, n (%)—no of observations (% of the group), Q1—first quantile, Q3—third quantile, SD—standard deviation; * Breastfeeding—means that the child from birth to at least 6 months of age was exclusively or predominantly breast-fed (in mixed feeding, no more than two portions of 150 mL of modified milk per day). This excludes children who took more formula per day or were breastfed for less than 6 months.

**Table 2 ijerph-20-04774-t002:** Univariate logistic regression model results for asthma risk within the cohort of children included in the study (N = 85).

Variable	OR	95% CI	*p*
Gender, female	0.87	0.37–2.03	0.744
Age, years	1.31	1.14–1.57	0.001
BMI, kg/m^2^	1.08	0.97–1.24	0.189
Birth weight, g	1.00	1.00–1.00	0.370
Mother’s age, years	1.12	1.00–1.27	0.060
Delivery type:			
Natural spontaneous	*RC*	-	-
Natural induced	1.79	0.43–7.94	0.420
Forceps or vacuum	8.61	2.09–58.79	0.008
Caesarean section	0.96	0.12–6.21	0.963
Breastfeeding *	0.04	0.00–0.21	0.002
Mother’s tertiary education	1.82	0.61–5.86	0.294
City residence	7.22	2.13–33.40	0.004
Village residence	0.13	0.03–0.44	0.003
Allergy in family	15.58	4.71–71.65	<0.001
Atopic dermatitis	2.70	0.88–9.33	0.093
Food allergy	2.37	0.76–8.29	0.148
AR/conjunctivitis	38.06	12.02–146.10	<0.001
Atopy	102.86	24.43–733.27	<0.001
Type of allergy			
HDM	14.67	4.43–67.36	<0.001
Molds	5.59	1.33–38.36	0.035
grass/trees pollen	24.32	8.11–87.00	<0.001
dog/cat	1.58	0.42–6.63	0.503
Nicotine exposure	0.73	0.22–2.31	0.592
Domestic animals	1.26	0.54–2.99	0.594
Dampness	1.62	0.64–4.18	0.312
Carpet surface	6.21	2.03–23.52	0.003
Cleaning:			
Less than once a week	0.00	-	0.991
Once a week	*RC*	-	-
Twice a week	1.20	0.40–3.54	0.736
More than twice a week	0.31	0.08–1.08	0.072
Use of antibiotics up to the age of 2	7.33	2.85–20.40	<0.001
Use of more than 2 antibiotics/year in childhood	6.40	2.53–17.35	<0.001
Immunological markers			
PPARG+/11c+	1.01	0.97–1.05	0.593
25+/4+	1.04	0.98–1.12	0.197
FOXP3+	0.95	0.93–0.97	<0.001
GARP+ (I)	1.00	0.98–1.01	0.599
25+/71+ (I)	0.96	0.89–1.02	0.270
SOCS+	1.01	0.99–1.03	0.190
25+/71+ (II)	0.97	0.91–1.01	0.232
GARP+ (II)	0.99	0.96–1.01	0.227
FEV1, L	3.51	1.61–9.10	0.004
FEV1, %	0.92	0.86–0.97	0.006
FEV1/FVC, %	0.001	0.000–0.131	0.009
FeNO, ppb	1.12	1.04–1.23	0.010

BMI—body mass index, AR—allergic rhinitis, HDM—house dust mites, PPARG+—Peroxisome proliferator-activated receptor gamma, FOXP3+—Forkhead box P3, GARP+—Glycoprotein A repetitions predominant, SOCS+—Suppressors of cytokine signaling, FEV1—forced expiratory volume in first second, FVC—forced vital capacity, FEV1/FVC ratio—also called Tiffeneau–Pinelli index, the ratio of the forced expiratory volume in one second to the full forced vital capacity, FeNO—fraction of exhaled nitric oxide, *RC*—reference category, OR—odds ratio, CI—confidence interval. * Breastfeeding—means that the child from birth to at least 6 months of age was exclusively or predominantly breast-fed (in mixed feeding, no more than two portions of 150 mL of modified milk per day). This excludes children who took more formula per day or were breastfed for less than 6 months.

**Table 3 ijerph-20-04774-t003:** Multivariate logistic regression for asthma risk within the cohort of children included in the study (N = 85).

Variable	OR	95% CI	*p*
Breastfeeding *	0.21	0.01–1.73	0.204
City residence	3.23	0.42–31.82	0.274
Other allergic disease **	41.12	5.77–896.52	0.002
Use of antibiotics up to the age of 2	7.14	1.49–43.90	0.019
FOXP3+	0.94	0.91–0.97	0.001

FOXP3+—Forkhead box P3, OR—odds ratio, CI—confidence interval. * Breastfeeding—means that the child from birth to at least 6 months of age was exclusively or predominantly breast-fed (in mixed feeding, no more than two portions of 150 mL of modified milk per day). This excludes children who took more formula per day or were breastfed for less than 6 months. ** At least one of: atopic dermatitis, food allergy, allergic rhinitis, atopy.

**Table 4 ijerph-20-04774-t004:** Level of FOXP3+ depending on selected environmental factors.

Variable	n	Me (Q1; Q3)	MD (95% CI)	*p*
Breastfeeding *				
Yes	68	49.60 (30.07; 75.17)	28.10 (8.30; 37.60)	0.001
No	17	21.50 (12.00; 33.60)		
Residence				
City	66	36.55 (21.85; 61.10)	−22.15 (−31.50; 1.00)	0.067
Village	19	58.70 (37.95; 89.00)		
Use of antibiotics up to the age of 2 **				
Yes	47	35.10 (21.70; 57.20)	−14.90 (−23.50; 3.00)	0.141
No	37	50.00 (31.60; 75.00)		
Use of antibiotics more than 2 per year in childhood **				
Yes	46	39.25 (21.65; 75.60)	−4.05 (−13.20; 12.80)	0.822
No	38	43.30 (26.65; 60.80)		
Other allergic disease ***				
Yes	57	32.70 (18.90; 59.00)	−25.90 (−34.90; −9.00)	0.001
No	28	58.60 (39.88; 91.38)		
Cleaning				
Once a week or less frequently ****	21	43.40 (22.00; 85.50)	-	0.043
Twice a week	42	34.90 (21.52; 56.10) ^a^		
More than twice a week	22	57.70 (34.48; 89.70) ^a^		

Me—median, Q1—first quantile, Q3—third quantile, MD—median difference (factor present vs. absent), CI—confidence interval. Difference between groups verified with: Mann-Whitney U test and Kruskal–Wallis test. a—significant difference based on post-hoc analysis (Dunn test with Bonferroni adjustment), *p* = 0.014. * Breastfeeding: “Yes”: means that the child from birth to at least 6 months of age was exclusively or predominantly breast-fed (in mixed feeding, no more than two portions of 150 mL of modified milk per day). “No” means exclusion from the subgroup of children described as breastfed because they were fed more formula per day than stated in the survey, or because they were breastfed for less than 6 months. ** Sum of observations lower than N = 85 due to data missing for one patient. *** At least one of: atopic dermatitis, food allergy, allergic rhinitis, atopy. **** “Once a week” and “less than once a week” were combined into one group due to low number of observations for “less than once a week” (n = 1).

**Table 5 ijerph-20-04774-t005:** ROC analysis results—assessment of predictive value of FOXP3 for asthma detection.

Cut-Off Point	AUC (95% CI)	Sensitivity	Specificity	Accuracy	PPV	NPV	*p*
33.65	0.821 (0.731;0.911)	0.67	0.86	0.76	0.82	0.73	<0.001

AUC—area under ROC curve, CI—confidence interval, PPV—positive predictive value, NPV—negative predictive value.

## Data Availability

Not applicable.

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
