# Peer review of "The Effect of Environmental Factors on Immunological Pathways of Asthma in Children of the Polish Mother and Child Cohort Study"

_ijerph, 2023, doi:10.3390/ijerph20064774_

Round 1

Reviewer 1 Report (Previous Reviewer 2)

The manuscript in it's current form has serious flaws in terms of the methodology. These flaws that were also addressed in the previous revision round should be adjusted: 

- The paper should mention in the discussion that they performed multiple tests in a low number of participants. Therefore there is a risk of finding false positive effects due to multiple testing. The current discussion does not give enough focus to this major limitation in the study design. 

- The covariates for multivariate logistic regression were deliberately chosen. There should at least be a directed acyclic graph to make sure that the right covariates were chosen and due to the low number of participants, covariates should not be left out based on the p-value. 

Author Response

Thank you , 

all responses are atttached below

Reviewer 2 Report (Previous Reviewer 3)

Thanks for the revision done.  It can be published

Author Response

Thank you 

Round 2

Reviewer 1 Report (Previous Reviewer 2)

Thank you for adjusting the paper as suggested. 

This manuscript is a resubmission of an earlier submission. The following is a list of the peer review reports and author responses from that submission.

Round 1

Reviewer 1 Report

I would like to congratulate the authors for the work done, I think it is a very interesting manuscript and that it provides new information that should be published. However, there are some aspects that could be improved prior to publication.

-          In the abstract section: Location of where the study was perfomed should be added (Poland)

-          Introduction section:

-           Add the proper references in the 1st paragraph of the manuscript.

-          Study design:How many patients were recruited in their first trimester of pregnancy?

-          Pg 3.lines 121- please clarify the asthma definition from the GINA guidelines and reference properly.

-          table 1, 2: please add footer of all the abbreviations

-          Table 2, table3, table 4. What is the * symbol means? Please specify

-          Results sections: Add the p or explaining if the results are significant or not could help to clarify the results.

-          Table 5: A sensitivity of 0,67 is not very high. How could you justify it?

Author Response

Dear Reviewer, thank you for your work. All suggested corrections have been made and described.

Q1: In the abstract section: Location of where the study was performed should be added (Poland)

Resp: The information concerning the place has been added as suggested. All corrections are marked in red.

Q2: Introduction section: Add the proper references in the 1st paragraph of the manuscript.

Resp: Thank you for that comment. The references were added.

Q3: Study design: How many patients were recruited in their first trimester of pregnancy?

Resp: “Briefly, in 2007-2011, 1.300 mother-child pairs were recruited during the first trimester of pregnancy at maternity units or clinics in two regions of Poland (Lodz and Legnica) if they fulfilled the following inclusion criteria: single pregnancy up to 12 weeks of gestation, no assisted conception, no pregnancy complications, and no chronic diseases as specified in the study protocol”. All corrections are marked in red in the text.

Q4: Pg 3. lines 121- please clarify the asthma definition from the GINA guidelines and reference properly.

Resp: Pediatric asthma was diagnosed by a specialist in accordance with the recommendations of GINA (Global Initiative for Asthma) [25], which defines asthma as a heterogeneous disease characterized by chronic inflammation of the airways, the symptoms of which include wheezing, coughing, shortness of breath, chest tightness. As reported by GINA, symptoms vary over time and severity, and that variable expiratory airflow limitation may coexist, which may become permanent over time. The diagnosis was additionally supported by pulmonary function tests, improvement after dedicated asthma treatment or symptoms indicative of bronchial hyperresponsiveness - a tendency to excessive contraction in response to stimuli such as physical or chemical factors that do not cause such an effect in a healthy person. This paragraph was added to the text.

Q5: table 1, 2: please add footer of all the abbreviations

Resp: The footer was added below the tables.  

BMI - body mass index, AR – allergic rhinitis, HDM – house dust mites, PPARG+ - Peroxisome proliferator-activated receptor gamma, FOXP3+ -Forkhead box P3, GARP+ - Glycoprotein A repetitions predominant, SOCS+ - Suppressors of cytokine signaling, FEV1 - forced expiratory volume in first second, FVC -  forced vital capacity, FEV1/FVC ratio - also called Tiffeneau-Pinelli index, the ratio of the forced expiratory volume in one second to the full forced vital capacity, FeNO - fraction of exhaled nitric oxide, n (%) – no of observations (% of the group), Q1 – first quantile, Q3 – third quantile, SD – standard deviation

-Q6: Table 2, table3, table 4. What is the * symbol means? Please specify

Resp:

Table 2  “Mother’s education -higher*”- * means: higher compared to secondary education.

Table 3  “Other allergic disease*”- * means: at least one of: atopic dermatitis, food allergy, allergic rhinitis, atopy

Table 4  

“Other allergic disease*”-  * means: at least one of: atopic dermatitis, food allergy, allergic rhinitis, atopy.

“Use of antibiotics up to 2. year** “   - ** means: sum of observations lower than N = 85 due to data missing for one patient.

Cleaning:  “Once a week or less frequently*** -  ***means; “Once a week” and “less than once a week” were combined into one group due to low number of observations for “less than once a week” (n = 1).

-Q7: Results sections: Add the p or explaining if the results are significant or not could help to clarify the results.   

Resp: All p-values are marked in the tables. However all significant p-values were also mentioned in the results section.

-Q8: Table 5: A sensitivity of 0,67 is not very high. How could you justify it?

Resp: The sensitivity of a diagnostic test describes the test's ability to detect individuals with a given trait. Low sensitivity may result from insufficiently large group size, however, this is not a big obstacle, because tests with high sensitivity are burdened with a higher risk of false positive results. There are no perfect tests in medicine, and with high specificity of the test it is difficult to achieve also very high sensitivity of the test.

English language and style were corrected by a native.

Reviewer 2 Report

Zywiolowska-Smuga et al. investigated which factors may be associated with increased risk of asthma during childhood. The statistical methodology used in the manuscript requires improvement. The following points need to be addressed: 

Major issues: 

Abstract

- The abstract is lacking a clear description of the aim of the research

- The abstract does not explain what FOXP3 is, please explain this shortly in the abstract. 

Methods

- Were the controls matched to the children with asthma based on certain factors? Please describe whether this is conducted or why this has not been done. 

- Please check the spelling of the statistical tests that were used to conduct the research. 

Results

- Table 1: the use of the p-value and MD/RR is not needed here in the description of the characteristics of the population. The groups made in this table are very small and the p-value do not reflect differences between patients with asthma and without asthma in a larger population. Please find further justification here: Sterne J, Davey-Smith G. Sifting the evidence — What’s wrong with significance tests? BMJ 2001; 322: 226-231. See also: Wasserstein RL, Lazar NA. The ASA’s statement on P-values: context, process, and purpose. The American Statistician 2016: DOI: 10.1080/00031305.2016.1154108.

- Table 2: there are a lot of analyses conducted here in a small number of patients (n=85). How do the authors correct for multiple testing? Were power calculations performed? This is not stated in the methods section. 

- Table 2 and results section: 2. year is confusing to the readers. Does this mean 2 times per year?

- paragraph 3.1.3: please add that the level of FOXP3 was found in the blood of the patients

- paragraph 3.2: please explain why these variables were chosen. Due to the multiple testing and the low number of patients, decision making based on p-values may not be reliable here. 

Discussion:

- Please describe here also the statistical pitfalls due to the low power of the current study design. 

Minor issues:

- Table 3: please change the comma into points according to the English grammar rules.

Author Response

Dear Reviewer, thank you for your work and comments.

All suggested corrections have been made and described.

Question 1: The abstract is lacking a clear description of the aim of the research

Response: "The aim of the study was to assess the influence of the environment on the development of asthma and its immunological markers.  We hypothesized that in our cohort, exposure to environmental factors is associated with asthma risk in children, and that FOXP3 levels vary with their incidence and are negatively correlated with developing asthma."  All corrections are marked in red in the manuscript.

Q2: The abstract does not explain what FOXP3 is, please explain this shortly in the abstract. 

Resp: The FOXP3 transcription factor is a marker of regulatory T cells (Tregs) and is essential in the process of their activation and proper expression, which promotes immune homeostasis both by enhancing the secretion of anti-inflammatory cytokines and by maintaining tolerance to self-antigens and induced Treg lymphocytes. We didn’t add this explanation in the abstract as according to the journal requirements the abstract is supposed to have up to 200 words.

Q3: Methods

Were the controls matched to the children with asthma based on certain factors? Please describe whether this is conducted or why this has not been done. 

Resp: The control group for the study was taken from the same cohort of patients who were not diagnosed with asthma or allergic diseases. Patients met the same criteria for inclusion in the study. The description has been shortened due to the need to fit within the imposed number of words.

Q4: Please check the spelling of the statistical tests that were used to conduct the research. 

Resp: The correct names of statistical tests should be as follows:  Student’s t-test, Mann Whitney U test, Fisher’s exact test, chi-squared test. All corrections are marked in red in the text.

Results

-Q5: Table 1: the use of the p-value and MD/RR is not needed here in the description of the characteristics of the population. The groups made in this table are very small and the p-value do not reflect differences between patients with asthma and without asthma in a larger population. Please find further justification here: Sterne J, Davey-Smith G. Sifting the evidence — What’s wrong with significance tests? BMJ 2001; 322: 226-231. See also: Wasserstein RL, Lazar NA. The ASA’s statement on P-values: context, process, and purpose. The American Statistician 2016: DOI: 10.1080/00031305.2016.1154108. –

Resp: Thank you for that comment. All p-values and MD/RR columns were removed from Table 1 as suggested due to the size of the groups and the impossibility of reflecting these values to a larger population.

Q6: Table 2: there are a lot of analyses conducted here in a small number of patients (n=85). How do the authors correct for multiple testing? Were power calculations performed? This is not stated in the methods section. 

Resp: A relatively small sample size admittedly tends to lower the statistical power. However, modern and computerized statistical analyses allow researchers to reach out for plausible non-parametric and/or exact tests which return reliable and confident estimates. Please consider that total of 85 study participants is not small.

Investigated traits had been tested for their normality and homogeneity of variances before main hypothesis testing. The Authors deliberately chose non-parametric tests with robust standard errors given the discussed not large sample size. The Authors did not estimate the statistical power. As regards multiple testing in n = 85, the Authors fitted Dunn or Mann-Whitney-Wilcoxon U tests in those situations, bearing in mind the sample size they were dealing with.

Q7: Table 2 and results section: 2. year is confusing to the readers. Does this mean 2 times per year?

Resp: Thank you for that comment. We have corrected for clarity as follows:

“Use of antibiotics up to the age of 2” and  “Use of more than 2 antibiotics per year in childhood”.

Q8: paragraph 3.1.3: please add that the level of FOXP3 was found in the blood of the patients

Resp: The corrections were made as suggested.

"FOXP3 levels were found in the blood of patients in both groups. Level of FOXP3 was negatively related to the risk of asthma, its level higher by one unit decreased the risk by 5%. No association was found between asthma and other immunological parameters we considered in our study.”

Q9: paragraph 3.2: please explain why these variables were chosen. Due to the multiple testing and the low number of patients, decision making based on p-values may not be reliable here. 

Resp: Due to the total number of observations, the dimensioned model had to be limited to five variables. The choice was dictated by the importance of the factors we wanted to assess. Since we are anxiously observing the increasing antibiotic resistance of many bacteria, we wanted to highlight one more important argument why it is worth making decisions about antibiotic therapy very carefully, especially in children, because we assumed that it may also have a negative impact on the development of asthma. In addition, we chose breastfeeding for the assessment to check whether the mother can provide anti-asthmatic protection in the first months of her child's life through natural feeding. The presence of the "other allergic diseases" factor is to assess whether in fact children in this age we observe symptoms of other allergic diseases may develop asthma more easily and therefore it is even more important for them to be protected against factors that increase the risk of asthma that we know and exposure to those that have a protective effect.

Discussion:

Q10: Please describe here also the statistical pitfalls due to the low power of the current study design.   

Possible low power of study design might lead to lower chance of detecting effects and relationships of small magnitude. This suggests that factors influencing asthma risk in a relatively weak way might have not been identified due to relatively limited sample size. Broader sample size would lead to higher power and, thus, larger likelihood of enriched and more detailed findings as well as it would confirm the findings of the current study. 

Q11: Table 3: please change the comma into points according to the English grammar rules. 

Resp: Corrected as suggested

English language and style were corrected by a native.

Thank you.

Reviewer 3 Report

The article is well structured and manages to give the meaning of the research in a clear way. The pathogenesis of childhood asthma development is an ever-evolving topic. This study is important because it is prospective that it is based on children's data from the Polish Mother and 71 Child Cohort (REPRO_PL). The study design is well done and allergen sensitization with skin prick testing was performed. The results are well illustrated. The study contributes to the agreement with other similar ones regarding FOXP3 levels and it offers an opportunity for future studies related to the airway tract microbiota and its influence on the pathogenesis of asthma in children. Indeed, In fact, my suggestions are given below.

- Minor issues

1) In the 4. Discussion after "... the mechanisms leading to the development of asthma in children ..." also a brief part about the microbiota of the upper respiratory tract and the development of asthma in children (see Santacroce L et al. The Human Respiratory System and its Microbiome at a Glimpse. Biology (Basel). 2020;9(10):318doi:10.3390/biology9100318 ).

2) In the introduction up to line 53 no they are references but they start later. Better to specify also in these lines.

Author Response

Dear Reviewer,

Thank you for your work and comments. All suggested corrections have been made and described.

Q1) In the 4. Discussion after "... the mechanisms leading to the development of asthma in children ..." also a brief part about the microbiota of the upper respiratory tract and the development of asthma in children (see Santacroce L et al. The Human Respiratory System and its Microbiome at a Glimpse. Biology (Basel). 2020;9(10):318doi:10.3390/biology9100318 ). 

Resp: Thank you for that comment. The article you mention enriches this work.

This information was added to the text in Discussion.

Q2) In the introduction up to line 53 no they are references but they start later. Better to specify also in these lines.

Resp: The references were corrected.

English language and style were corrected by a native.

Thank you

Round 2

Reviewer 1 Report

After major revisions it has improved,  I think it should be published.

Author Response

Thank you for your comments making the work have a chance to be better. 
I have applied all your comments to the changes to the Manuscript. Changes and clarifications made to the Manuscript in the second round are in red and underlined in yellow.
I attach the answers to the comments I received from the Academic Editor in the second round in a separate file.